# Experimental evidence for the existence of a second partially-ordered phase of ice VI

Ryo Yamane [1,2✉], Kazuki Komatsu [1], Jun Gouchi[2], Yoshiya Uwatoko [2], Shinichi Machida[3], Takanori Hattori[4], Hayate Ito[1] & Hiroyuki Kagi[1]

Ice exhibits extraordinary structural variety in its polymorphic structures. The existence of a new form of diversity in ice polymorphism has recently been debated in both experimental and theoretical studies, questioning whether hydrogen-disordered ice can transform into multiple hydrogen-ordered phases, contrary to the known one-to-one correspondence between disordered ice and its ordered phase. Here, we report a high-pressure phase, ice XIX, which is a second hydrogen-partially-ordered phase of ice VI. We demonstrate that disordered ice undergoes different manners of hydrogen ordering, which are thermodynamically controlled by pressure in the case of ice VI. Such multiplicity can appear in all disordered ice, and it widely provides a research approach to deepen our knowledge, for example of the crucial issues of ice: the centrosymmetry of hydrogen-ordered configurations and potentially induced (anti-)ferroelectricity. Ultimately, this research opens up the possibility of completing the phase diagram of ice.

[1] Geochemical Research Center, Graduate School of Science, The University of Tokyo, 7-3-1 Hongo, Bunkyo-ku, Tokyo, Japan. [2] The Institute for Solid State Physics, The University of Tokyo, 5-1-5 Kashiwanoha, Kashiwa, Chiba, Japan. [3] Neutron Science and Technology Center, CROSS, 162-1 Shirakata, Tokai, Naka, Ibaraki, Japan. [4] J-PARC Center, Japan Atomic Energy Agency, 2-4 Shirakata, Tokai, Naka, Ibaraki, Japan. ✉email: r.yamane@issp.u-tokyo.ac.jp

To date, more than 20 crystalline and amorphous phases of ice have been reported[1]. This extraordinary polymorphism makes ice unique with its universality and has inspired many studies in the wide-ranging fields of material science and Earth and planetary science. The structural variety of ice arises from the geometric flexibility of hydrogen bonds and hydrogen order-disorder phase transition[2], and hydrogen ordering in ice structure also induces significant changes in the dynamic/static properties of ice along with the phase transition, such as immobilisation of molecular rotation[3] and ferro- or antiferroelectrically aligned molecular structures[4–6]. A prominent unsolved question[7] concerning the structural diversity induced by hydrogen ordering is whether a hydrogen-disordered phase of ice transforms into only one hydrogen-ordered phase, as inferred from the currently known phase diagram of ice, although the energies of its possible hydrogen configurations are close because of the inherent geometric frustration of the ice lattice[8–12]. Recent experiments on a high-pressure hydrogen-disordered phase, ice VI, revealed an unknown hydrogen-ordered form (β-XV[13]) besides the known ordered phase, ice XV[14]. Although the unknown ordered form would be a counterexample of the question, it has not been clarified whether β-XV is a distinct crystalline phase to be assigned as a new Roman numeral label due to lack of experimental evidence[7,13,15–17]. Herein we report a second hydrogen-ordered phase for ice VI, ice XIX, unambiguously demonstrated by in-situ dielectric and neutron diffraction measurements under high pressure. The phase boundary between ice VI and ice XIX shows that ice VI contracts upon hydrogen ordering, which thermodynamically stabilizes ice XIX in the higher-pressure region compared to ice XV because of the smaller volume of ice XIX than ice XV[8,14,18]. The pressure-induced multiplicity of hydrogen-ordered phases, also theoretically suggested in other ice polymorphs[11], can induce (or control) the different manners of hydrogen ordering of ice. Thus, this study demonstrates a hitherto undiscovered polymorphism of ice.

## Results

### Dielectric measurements of ice VI, XV, and XIX.
Comprehensive observation of hydrogen ordering in ice VI was conducted by dielectric experiments in the pressure range 0.88–2.2 GPa. Ice VI was initially obtained at room temperature and its dielectric properties were determined in both cooling and heating runs in the temperature range 100–150 K, using a newly developed pressure cell (see Methods and Supplementary Fig. 10). After the heating runs, the sample was subsequently heated to room temperature for annealing. Then, the sample was compressed again, and dielectric measurements were conducted at different pressures (Fig. 1).

Phase transitions from ice VI to its hydrogen-ordered phases were observed at around 120–130 K, along with sudden weakening of the dielectric response of ice VI with decreasing temperature (Fig. 2). The reason for this weakening is that hydrogen ordering of ice suppresses reorientation of water molecules which induces the dielectric response of ice[2,19]. The hydrogen-ordering is also supported by the temperature dependence of the relaxation time of ice VI described in Supplementary Fig. 3, where the relaxation time of ice VI becomes longer along with the hydrogen ordering because of the suppression of molecular reorientation. We herein determined the phase-transition temperatures from ice VI to its hydrogen-ordered phases based on the sudden weakening of dielectric loss peak intensity accompanied with the phase transitions (Fig. 2b). In the determination, we used an assumption that the peak intensity of ice VI has a linear relationship with temperature (see detailed discussion in Supplementary Method 2.2). The slope of the

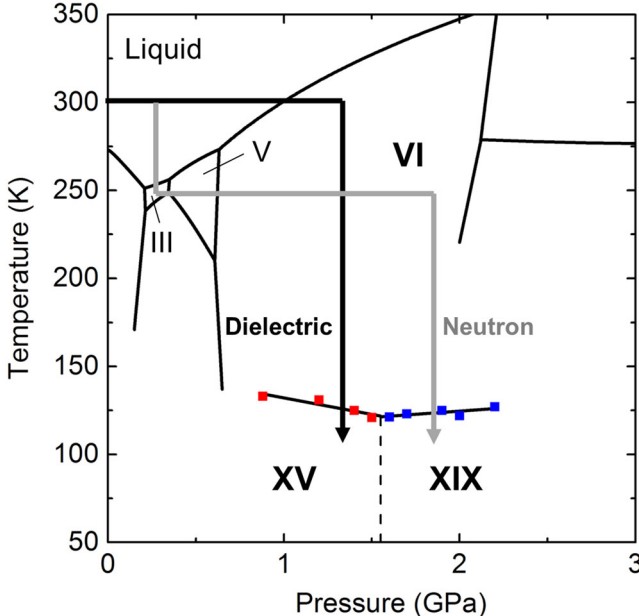

**Fig. 1 Representative experimental paths of dielectric and neutron diffraction experiments described in the phase diagram of ice obtained herein.** Dielectric experiments of ice VI and its hydrogen-ordered phases were conducted at 0.88–2.2 GPa. HCl (99.9%, Wako) was introduced as a dopant (concentration: $10^{-2}$ M) to accelerate the hydrogen ordering of ice VI[12]. The measured temperature was in the range 100–150 K and changed at a rate of 2 K/h for all dielectric measurements. Neutron diffraction experiments of DCl-doped $D_2O$ (concentration: $10^{-2}$ M) were conducted using a more complicated path to ensure that the sample was a fine powder through solid–solid phase transitions, i.e., ice III → ice V → ice VI. Sample diffraction was collected at 1.6 and 2.2 GPa, and the temperature range was 80–150 K. Temperature was changed at a rate of 6 K/h. Diffraction patterns were collected using new samples in each run at different pressures to confirm reproducibility. Phase boundaries among ice VI, ice XV, and ice XIX are described by black solid lines, based on dielectric experiments (red and blue squares correspond to phase transition temperatures from ice VI to ice XV and XIX, respectively). The dotted line shows the provisional phase boundary between ice XV and ice XIX (see main text).

obtained phase boundary, i.e. $dT/dP$, between ice VI and its hydrogen-ordered phases changes from negative to positive at 1.5–1.6 GPa with increasing pressure (Fig. 1). Based on the Clausius–Clapeyron relationship, i.e., $dT/dP = \Delta V/\Delta S$, this sign change for $dT/dP$ strongly indicates that ice VI has two different hydrogen-ordered phases with opposite signs for $\Delta V$, because $\Delta S < 0$ generally holds for hydrogen ordering. Since the currently known hydrogen ordering from ice VI to ice XV shows a positive volume change (observed at the lower pressure, 0.4 GPa[8]), ice XV is in the lower pressure region and the hydrogen-ordered phase in the higher-pressure region is a new phase, ice XIX, which has a smaller volume than ice VI and also ice XV. The appearance of ice XIX is governed by the $PV$ term in the Gibbs energy expression, because the volume contraction thermodynamically stabilises ice XIX compared to ice XV. In this context, the phase boundary between ice XV and XIX would be like close to vertical unlike horizontal suggested in the previous study[13] if their entropy difference is enough small, because ice XV has a larger volume than ice XIX (the supposed phase boundary in Fig. 1 is shown vertically to emphasise this point). It is noteworthy that the phase transition between ice VI and XIX showed hysteresis for the transition temperature (Supplementary Fig. 1). This first-order phase transition is consistent with the sudden change in dielectric properties between ice VI and ice XIX (Fig. 2).

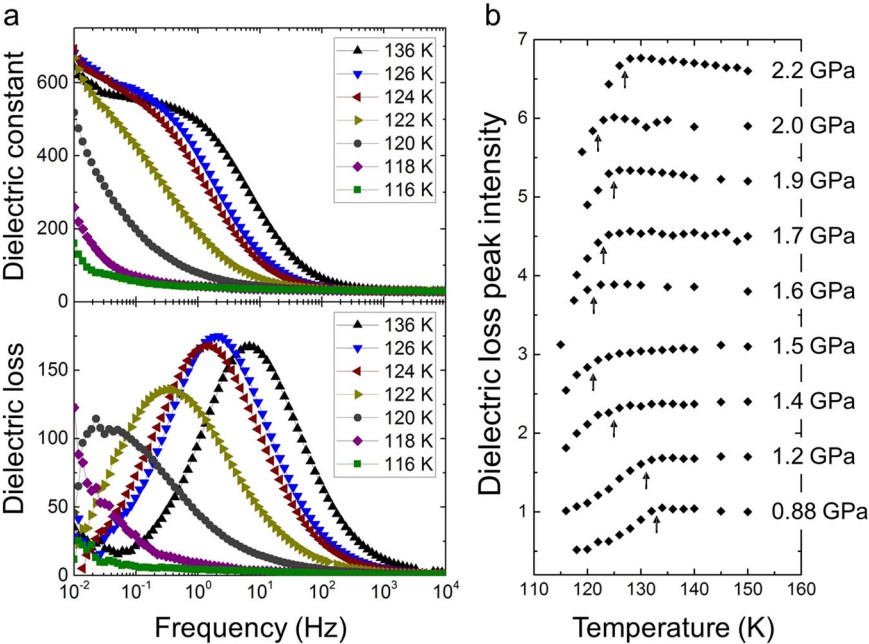

**Fig. 2 Temperature dependence of dielectric properties of HCl-doped ice VI and its hydrogen-ordered phases. a** Dielectric constant and dielectric loss of HCl-doped ice VI and its hydrogen-ordered phase (ice XIX) obtained at 1.9 GPa upon cooling using an LCR meter (NF corp., ZM2371). The measured frequency was from 3 mHz to 2 MHz. **b** Temperature dependence of dielectric loss peak intensity of HCl-doped ice VI and its hydrogen-ordered phases obtained in the pressure range 0.88–2.2 GPa upon cooling (black diamonds). Each peak intensity of dielectric loss was estimated using a model fitting for the corresponding dielectric loss spectrum based on the Debye dielectric-relaxation equation (polydispersion type). Under each pressure, peak intensities were normalised by that obtained at the highest temperature. Each plot was shifted by 0.7 with increasing pressure for clarity. In each pressure, the transition temperature whose determination is described in Supplementary Method 2.2 is indicated by black arrows.

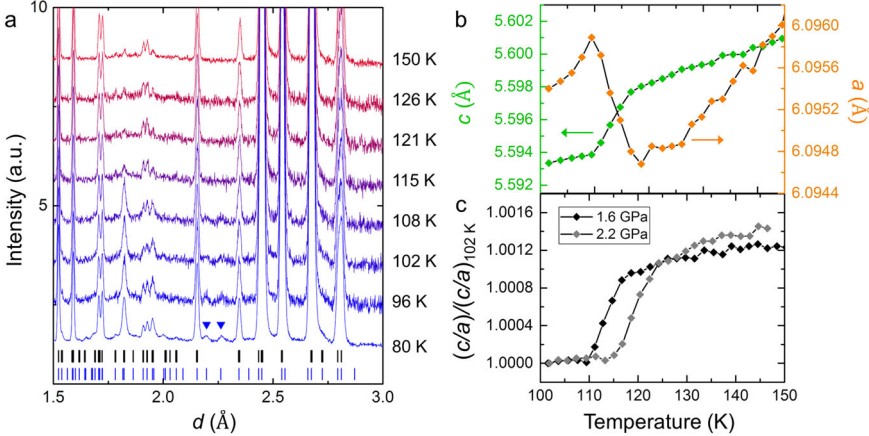

**Fig. 3 Temperature dependence of neutron diffraction patterns and lattice parameters of DCl-doped $D_2O$ ice VI and ice XIX. a** Neutron diffraction patterns of ice VI and XIX obtained at 1.6 GPa in the cooling run. Only an expanded area showing new peaks of ice XIX is displayed. The blue and black ticks represent all the peak positions expected from the unit cells of ice XIX and ice XV, respectively. Blue triangles indicate new peaks at 2.20 Å and 2.26 Å, which do not appear from the unit cell of ice XV. **b** Temperature dependence of lattice parameters, $a$ and $c$, of ice VI or ice XIX obtained at 1.6 GPa. The values were calculated based on the ice VI structure model even for ice XIX because of a common oxygen framework between ice VI and XIX. **c** Temperature dependence of $c/a$ at two different pressures, 1.6 and 2.2 GPa, indicated by black and grey. Phase transition from ice VI to ice XIX started at around 117 K and 124 K in the respective cooling runs. The $c/a$ values are normalised by that at 102 K. Diffraction patterns were collected using new samples in each run under different pressures to confirm reproducibility.

**Neutron diffraction measurements of ice VI and XIX.** Neutron diffraction experiments were conducted at 1.6 and 2.2 GPa to confirm whether ice XIX is a hydrogen-ordered crystalline phase distinct from ice XV. Both cooling and heating runs were conducted at each pressure in the temperature range 80–150 K.

A transition from ice VI to ice XIX was also observed in the neutron diffraction experiments, as appearance of new peaks due to symmetry lowering (Fig. 3a). Some of the new peaks, e.g. those

at 2.20 Å and 2.26 Å (indicated by blue triangles in Fig. 3a), cannot be assigned to the unit cell of ice XV; instead, they can be assigned to an expanded $\sqrt{2} \times \sqrt{2} \times 1$ cell with respect to the unit cell of ice VI (the unit cell of ice XV has a $1 \times 1 \times 1$ cell with respect to that of ice VI). This is unambiguous evidence that the hydrogen-ordered phase found in the higher-pressure region is a crystalline phase distinct from ice XV, and that ice VI has two different types of hydrogen ordering (neutron diffraction patterns

collected at 2.2 GPa also show the new peaks; see Supplementary Fig. 5). The reflection conditions show that the unit cell of ice XIX has a primitive lattice. The reduced unit cell parameters of ice XIX, $a$ and $c$, corresponding to the unit cell of ice VI, are expanded and contracted, respectively, upon hydrogen ordering (Fig. 3b); this tendency was also observed at 2.2 GPa. A comparison of the temperature dependences of $c/a$ at 1.6 and 2.2 GPa (Fig. 3c) showed that the phase-transition temperature at 2.2 GPa was at about 7 K higher than that at 1.6 GPa. This result is consistent with the phase boundary between ice VI and XIX obtained by the dielectric experiments. On the other hand, no significant volume change was observed in our neutron diffraction experiments, in contrast to the expected negative volume change ($\Delta V < 0$) upon hydrogen ordering, probably due to the small volume contraction.

**Structure analysis of ice XIX**. For the structure analysis of ice XIX, we considered candidates of its space group based on the group–subgroup relationship between ice VI and XIX, in addition to the experimentally confirmed reflection conditions. There are 36 subgroups for the space group of ice VI, $P4_2/nmc$, considering the primitive unit cell of ice XIX. Among them, thirteen space groups, having $h0l : h + l = 2n$ and $0kl : k + l = 2n$ reflection conditions, can be excluded from the observed reflection conditions. We conducted Rietveld analyses using structural models with 18 space groups of the remaining candidates, except for $Pc$, $P2_1$, $P2$, $P\bar{1}$ and $P1$, which are the lower-symmetry space groups of $P\bar{4}$, $Pca2_1$, $Pcc2$, $P2_1/a$ and $P2_1/c$, selected as the best candidates for ice XIX based on their fitting $\chi^2$ values. Since the candidates show close values of $\chi^2$ (between 5.4 and 6.4 as shown in Supplementary Fig. 8) and they can explain all observed peaks of ice XIX (see Supplementary Fig. 9), the lower-symmetry space groups were not considered here. Notably, we do not rule out the possibility that the actual crystal structure of ice XIX having one of the lower-symmetry space groups. A structural model of each candidate was constructed using a partially ordered model adopted in an earlier study[8]. The $P\bar{4}$ or $Pcc2$ structural models are the most plausible for the space group of ice XIX, based on the structure refinements. Considering the suggested space group of ice XV, $P\bar{1}$[14] or $Pmmn$[8], centrosymmetry of hydrogen configurations is the most significance difference in hydrogen configuration between ice XIX and ice XV. In particular, $Pcc2$ suggests a pyroelectric structure as well as ice XI and its polar direction is along the $c$ axis. Although further investigations, such as a single-crystal neutron diffraction experiment, are necessary to precisely determine the hydrogen configurations, centrosymmetry will be an intriguing point in structural studies of ice XV and XIX. In addition, the $P\bar{4}$ and $Pcc2$ structural models include deuterium (hydrogen) atoms whose site occupancy is 50 %; in other words, the models exhibit that ice XIX is partially ordered state as with the $Pmmn$ structure model suggested for ice XV by Komatsu et al[8]. The space groups ($P\bar{4}$ or $Pcc2$) indicate a possibility that there exists another fully hydrogen-ordered phase of ice VI instead of the partially hydrogen-ordered ice XIX considering the third law of thermodynamics. It is noted that Gasser et al. also came to the identical conclusion from their neutron diffraction measurements using decompressed samples at about the same time as we did. They referred our structural analysis and reported that the best fitting was obtained by the $P\bar{4}$ and $Pcc2$ structural models, the same as our results[20]. They reported volume contraction of the hydrogen ordering from ice VI to XIX at ambient pressure. This result supports the phase diagram of ice VI and its hydrogen ordered phases clarified in this study.

This study demonstrates the existence of multiple hydrogen-ordered phases for a hydrogen-disordered phase and clarifies the

effectiveness of applying pressure to induce phase competition among the hydrogen-ordered phases. Based on previous theoretical studies[8–11] and the currently known phase diagram of ice, the low-temperature region of the phase diagram (below approx. 150 K) is a frontier region for exploring undiscovered ways of hydrogen-ordering in ice, which would greatly change the phase diagram of ice. It is additionally noteworthy that the unit cell size of ice XIX allows many possible hydrogen-ordered configurations (1964 symmetry-independent configurations), such that an exhaustive theoretical analysis for the all configurations is difficult. However, such a wide variety of hydrogen-ordered configurations and their stability evaluations might be a good benchmark for modern theoretical trials toward modelling biochemical and environmental processes with a large number of water molecules, such as using topological graph invariant theory[21], combining oriented graph theory and density functional calculations, which can evaluate the energy stability of a large number of water-molecule arrangements. To the best of our knowledge, this is the first report in a hydrogen-bonded material for which different hydrogen-ordered configurations are realised depending on the pressure, although electric field is a known effective parameter to control ferro- and antiferroelectric structures of organic compounds[22,23]. This newly discovered coupling between hydrogen bond and pressure will extensively develop an research field of the pressure-controllability of hydrogen-ordered configurations. Furthermore, by combining high-electric field with high-pressure, this multi-extreme condition is expected to provide more various types of hydrogen ordering and physical properties (e.g., (anti-)ferroelectricity) to hydrogen-bonded materials. Very recent technical development of neutron diffraction experiment[24] will encourage the intriguing exploration in $P$-$T$-$E$ phase diagram focusing on the multiplicity of hydrogen-ordered phases.

## Methods

**Dielectric measurements**. We conducted in-situ dielectric measurements under high pressure using a newly developed cell assembly. One of the most notable features of our development is that along with measuring the dielectric properties of the sample, the sample pressure can be simultaneously estimated using the ruby fluorescence method. This feature allows us to closely investigate the phase structure of ice in terms of its hydrogen ordering. The cell assembly is based on a piston-cylinder-type high-pressure apparatus (see details of the cell assembly in Supplementary Fig. 10). In the dielectric experiments involving ice VI, HCl (99.9%, Wako) was introduced as a dopant (concentration: $10^{-2}$ M) to accelerate the hydrogen ordering of ice VI[12], and dielectric experiments were conducted on DCl-doped $D_2O$ ice VI (DCl concentration: $10^{-2}$ M) following the same experiment procedure as that for HCl-doped ice VI (Fig. 1). Pressure dependence of the phase transitions was similar to that of the HCl-doped $H_2O$ sample (see Supplementary Fig. 2). It is noted that sample pressure is slightly changed with decreasing (increasing) temperature at most 0.1 GPa in the measured temperature region. The shown pressure of the dielectric data corresponds to that measured at around phase-transition temperature. In the neutron diffraction measurement, the sample pressure also changed about 0.1 GPa in the measured temperature range, and the shown pressure was determined in the same manner as the dielectric measurements.

**Neutron diffraction measurements**. The neutron diffraction measurements were conducted at PLANET beamline 11 at the Materials and Life Science Experimental Facility of J-PARC, Ibaraki, Japan[25]. DCl-doped $D_2O$ sample was used as a starting material (DCl concentration: $10^{-2}$ M), and ice VI was prepared through solid–solid phase transitions, ice III → V → VI, to obtain a fine powder sample (Fig. 1). Pressure and temperature were controlled by using a Mito-system[26], and the pressure was estimated from the lattice parameter of Pb, which was added to the sample as a pressure marker[27].

**Structure analysis**. Initial candidates of ice XIX were determined based on the group–subgroup relationship between ice VI and XIX, using the SUBGROUPS program opened on the Bilbao crystallographic server[28–30]. The structural models for the 18 space group candidates were constructed based on the partially hydrogen-ordered model adopted in a previous study[8]. Hydrogen occupancies and atomic coordinates were the fitting parameters obeying the ice rule in this model (details of the 18 structural models are given in Supplementary Fig. 16 and

Supplementary Table 5). Structure refinements were conducted for the neutron diffraction patterns corrected at 1.6 GPa and 80 K using all the structure models employing the Rietveld method (program: GSAS with EXPGUI[31,32]). We used the initial structural parameters with the ice VI structural model, based on the neutron diffraction pattern obtained at 1.6 GPa and 80 K. It was noted that site occupancies of hydrogen atoms were initially varied in the refinements with fixed atomic positions of hydrogen. $\chi^2$ values of the structure models are plotted in Supplementary Fig. 6, where numerical values are shown only for five candidates with $\chi^2 < 9$. In the first step of structure refinements, the site occupancies of hydrogen atoms were fitted one by one, and subsequently fitted together as variables. Since the first step has arbitrariness in its fitting order (e.g. $\alpha \to \beta \to \cdots$ and $\beta \to \alpha \to \cdots$), we conducted structure refinements in several ways for each model by changing the fitting order cyclically, such as $\alpha \to \beta \to \cdots$ and $\beta \to \gamma \to \cdots$. However, the fitting results are almost independent of the order. The space group $P\bar{4}$ was deemed the most plausible candidate in this step. Supplementary Fig. 7 shows fitted lines using the five possible structural models (red-coloured) for the neutron diffraction pattern obtained at 1.6 GPa and 80 K (black lines). It should be mentioned that an observed Bragg peak, marked by a black tick at 1.82 Å, shows broadening compared to the simulated ones. This peak is derived from the new hydrogen-ordered phase. Generally, peak broadening arises from two factors, insufficient crystallite size and/or microstrain in the crystal, consistent with the partially hydrogen-ordered state of the ice XIX. Structure refinements, including atomic positions of hydrogen, were conducted for the five candidates (Supplementary Fig. 8); the fitted results obtained employing only hydrogen occupancies as fitting parameters were subsequently used for these refinements. In addition, O-D bond lengths were constrained to be 0.95, which is an averaged value of the O-D bond length of the ice VI structural model, based on the neutron diffraction pattern obtained at 1.6 GPa and 80 K. $\chi^2$ values of $P\bar{4}$ and $Pcc2$ were comparable considering the dispersion of their refinement results, and those structural models were considered the most plausible for ice XIX. Atomic fractional coordinates and O-D bond lengths for ice XIX using the two models are shown in Supplementary Tables 1–4, where lattice parameters have also been refined. Also, the fitted results for the neutron diffraction pattern collected at 1.6 GPa and 80 K are shown in Supplementary Fig. 9 regarding the two structure models.

## Data availability

The primary data that support the plots within this paper and other finding of this study are available from the corresponding author on reasonable request. The neutron crystallographic coordinates of ice XIX using $P\bar{4}$ and $Pcc2$ structure models have been deposited at the Cambridge Crystallographic Data Centre (CCDC), under deposition numbers 2045352 and 2045353, respectively. These data can be obtained free of charge from The Cambridge Crystallographic Data Centre via www.ccdc.cam.ac.uk/data_request/cif.

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

## Acknowledgements

We are grateful to the technical staff of the University of Tokyo (Graduate School of Science), Mr. S. Otsuka and Mr. T. Shimozawa, for their support in the experiments. Neutron diffraction experiments were performed using the J-PARC user program (proposal number 2019A0310). This research was supported by JSPS KAKENHI (Grant numbers: 19H00648, 18J13298, 18H05224, 18H01936, 15H05829).

## Author contributions

R.Y. conceived and designed the experiments. R.Y., J.G., and Y.U. developed the high-pressure cell for dielectric measurements. R.Y., J.G., and H.I conducted the dielectric experiments. R.Y., K.K., S.M., and T.H. conducted the neutron diffraction experiments. R.Y. and K.K. analysed the neutron diffraction data. R.Y. wrote the manuscript with contributions from K.K., T.H., and H.K. All the authors have discussed the data interpretation.

## Competing interests

The authors declare no competing interests.
