## [Peer Review File · Nature Communications]

REVIEWER COMMENTS

Reviewer #1 (Remarks to the Author):

Yamane et al report some measurements aimed at unambiguously establishing the presence and nature of a new, ordered form of ice VI that is distinct from the recently discovered ice XV and, which the authors name ice XIX. This result would be important as it would represent the first time such an observation has been made, suggesting that the phase diagram of ice could be even more complicated than is presently understood. Furthermore, using an analysis of neutron-diffraction data, they propose a structure for ice XIX that is, in itself, interesting due to centrosymmetry of the hydrogen configurations.

In light of the broad interest in new discoveries of phases of ice and of new hydrogen bonding phenomena, I believe this work would be of interest to the Nature Communications readership. I also believe that some persuasive evidence is presented. However, I had some significant concerns with the analysis used and interpretations drawn. I present these in detail below and would respectfully ask that the authors address them carefully before I could recommend publication.

One of the main claims of the paper is that ice XIX is denser than ice VI. However, I have a concern as to the robustness of this conclusion. Indeed, the authors own neutron diffraction-measurements were unable to measure any change in density. It is stated that this is due to the volume change being small, but exactly how small (i.e. how different it is from zero) was not qualified. It follows that the statement of negative “delta-V” at the VI-XIX transition hangs solely on the positive nature of the slope of the transition line (and the implication of the Clausius-Clapeyron relation), this slope being determined by fits to dielectric loss peak intensity (DLPI).

The DLPI data, as a function of pressure and temperature are shown in their Fig 2b and the resultant transition temperatures in Fig 1 for hydrogenous ice and HCl dopant. The first query I had relates to the colouring of the markers indicating the transition temperatures in Fig 1 and designating the phase: what determines which phase is which? If I remove the colour and ignore the drawn phase boundaries, then I could easily draw different lines through these data points, for example, the one shown in red here:

which seems similarly consistent with the data within the scatter other than, perhaps, the point at 1.6 GPa. In relation to the 1.6 GPa data point, I also have concerns with how the transition temperatures are extracted: by the intersection of two straight line fitted,

respectively, to the upper and lower temperature values of the DLPI at a given pressure. Especially in the case of the 1.6 GPa point this appears to artificially lower the transition temperature. If this single point is questionable, then it seems to significantly reduce the evidence for a positive slope for the transition temperature in the proposed XIX-VI region. If I were instead to consider fitting a straight line to the high temperature data, only using points in a region well away from the transition, then I could get quite a different behaviour. In the case of the 1.6 GPa data point, I might then put the transition point at the red arrow below and leading to a transition temperature that is more like the red-dashed line:

This would lead to a quite different interpretation, where there is a phase boundary around 1.4 GPa (incidentally, coinciding closely with the apparent transition temperature reported by Glasser et al of 1.45 GPa), then a horizontal transition temperature (consistent with the

authors observation of no volume change from diffraction) up to the last data point at 2.2 GPa at which, clearly, the transition temperature changes. However, could not the change of the 2.2 GPa data point be related to the approach to an ice VIII boundary?

In summary, on this point, although I don't claim my interpretation is more valid than the authors, I believe it illustrates a weakness in their volume argument and the location of the transition. In particular, the data should be presented with a serious analysis of systematic and random errors that could affect the transition temperatures extracted in this way. The proposal of a transition and, especially, the volume change have to be critically judged in light of this.

I then wanted to comment on the neutron data. The measurements were conducted along isobars at 1.6 and 2.2 GPa. *c/a* data are shown at both pressures, but the diffraction data are only shown at 1.6 GPa. The authors point to the appearance of weak, but to me unambiguous, peaks that appear in the diffraction pattern following a reduction of *c/a* ratio. This is quite convincing evidence of an ordering transition. As the new peaks are unindexed on ice VX, it does suggest that a new phase has been formed. I would have liked to see the data at 2.2 GPa and it's not clear why this doesn't appear in either the main manuscript or the supplementary material. It would surely be important to show that the same peaks were observed as at 1.6 GPa? I would also have been interested to see data at the same temperature, but lower pressure: the absence of these new peaks would strengthen the evidence for the location of the phase transition. This would be helpful especially as the DLPI data for the D2O sample seems less clear than for H2O.

I noted that in the neutron data, the new peaks appear between 108 and 115 K, but this seems to be cooler than the transition temperatures given by the DLPI. I wondered if this was an isotope effect (the D2O DLPI data seem less reliable than the H2O so hard to say), or whether it might also be an effect of cooling rate?

The authors conducted a quite thorough investigation of possible model fits to the neutron data and this is perhaps the most valuable work in the paper. I noticed that the consistent misfits of the peak width at $\sim 1.81 \text{ \AA}$ seem to be due to a large, Lorentzian-like broadening of the measured peak. Since such a peak-shape is often associated with particle-size effects, I wondered if this could be due to finite domain sizes of the ordered phase. In this case, they would be quite directional as only this peak is affected. Maybe this is worth further comment.

Lastly, on the refinements, there were several places where it was mentioned that the refinements were "conducted several times for each model" and the subsequent χ^2 values "averaged values over several refinement results". This seemed an unusual process to me, as I would expect a given model to converge to the exact same structure when refined to the same data. If it is true that the model converges to different local minima upon successive refinement cycles, it implies to me that something may be wrong and parameters in the refinement are unstable. I would wish the authors to comment further on this and to reassure me that the data are not being 'over fitted' with more parameters than available data points. In the case that data are being over fitted, the selection of a model, on the basis of it giving the lowest χ^2 may not be robust. It would have been helpful in the review to

have had a cif file and a secondary check on the model would have been whether sensible O-D bondlengths were obtained (coordinates are given, but I'm afraid I didn't have time to enter all of these manually). Also, were these freely refined or were constraints used?

Beyond the discussions above, I had several minor further queries, listed here:

line 90: "the phase boundary between ice XV and XIX should have a slope rather than lie horizontally as suggested previously". I found this confusing as the phase diagram the authors use has pressure on the x-axis and temperature on the y-axis. In this case, the transition between XV and XIX should be approximately *vertical* not horizontal?

line 156 and 162 attempt to reconcile DSC measurements on recovered samples, interpreted in terms of a "deep-glassy state of ice VI" with their observation of a new, ordered, crystalline phase. The authors seem to suggest that their ordered phase may disorder upon pressure lowering. This, to me, seemed a little counter intuitive: even if hydrogen mobility increases as pressure reduces, where would the additional energy come from to disorder the dipoles, which are already in a lower energy ordered state? The authors should provide some further justification for their explanation.

line 183 mentions "large water molecules" what does this refer to? Aren't all water molecules the same size?

Supplementary line 99: From the description of the pressure measurement, via ruby fluorescence, it seemed quite plausible to me that there would be a systematic deviation between ruby pressure and sample pressure, due to pressure gradients and that the ruby lies at one end of the sample. Given that the uncertainty in transition pressures of a few kbars would have a significant effect on the drawing of phase boundaries, did the authors attempt to quantify this and (if necessary) apply a correction?

Reviewer #2 (Remarks to the Author):

For the first time, a new hydrogen-ordered ice-phase has been described which shows a different hydrogen-ordering as compared to the already known hydrogen-ordered phase (ice XV) relating to the same known hydrogen-disorders phase (ice VI). The discovery, however, of such a phase belongs to Gasser et al., having reported on a beta-ice XV in 2018.

However, here is shown the first, unambiguous, structural evidence (by neutron diffraction) of such a phase.

Beside this, the paper presents a new technical development of a pressure cell allowing dielectric measurements.

Dielectric measurements delivered the first hints for a second, high-pressure form XIX in the phase boundary region of ice VI, XV and XIX. I fully back these results.

Then, neutron powder diffraction (avoiding formation of single crystals by a well-chosen pressure-temperature path) shall establish the experimental basis of an unambiguous structure analysis, showing a different hydrogen (well, deuterium) order as compared to ice XV. Structure solution was performed using group-subgroup relationships to find the potential ordering models to be tested. Out of the 18 possible subgroups, five were able to explain the observed diffraction patterns, and two of them have been retained as possible structure models.

Here comes my criticism: The selection was seemingly been done by comparison of χ^2 values as a criterion for the goodness of fit. The differences are not dramatic, as the supplementary figure 4 shows on the five best candidates. The final two candidates have been selected from these two only on the numerical χ^2 argumentation. This is, unfortunately, not satisfying, although I must admit that one has barely any other choice. All structure models seem to explain all observed peaks (except the retained Pcc2 one, where the peak at about $d=2.28\text{\AA}$ seems not to be explained ...). There's no further indication that the finally selected models are truly the good ones.

On the other side, the structure of XV is not unambiguous neither -- and it (seems to have) has different possibly symmetries as compared to ice XIX (not only the two retained models, also the five best ones, even the 18 considerable ones). This shall be valid as structural proof for ice XIX being of different structure as compared to ice XV, thus, a new phase.

So this work does NOT reveal the structure of ice XIX, it just confirms anew phase by conclusive dielectric measurements involving a new technical device development and structure analysis on neutron powder diffraction. The structure remains clearly unknown, but it seems to be clearly different from any potential ice XV structure. This discovery is sufficiently important to be published in a science journal read by many communities (geosciences, physics, physical chemistry, crystallography...), but it remains somehow unsatisfying. I know how difficult it is to get better quality neutron diffraction data from high pressure devices, and how difficult it will be to get even a single crystal, therefore I cannot ask the authors for much additional information to be incorporated right now.

Despite my concerns, I suggest the paper for publication. I do not insist on rephrasing, although some phrasing could have been clearer (especially to avoid the final deception at the end, when the interested reader does not find the expected unambiguous structure solution).

Reviewer #3 (Remarks to the Author):

This paper reports on experiments on H₂O ice in the range 0.88-2 GPa and low temperatures using dielectric and neutron diffraction methods, and claims the discovery of a new hydrogen-ordered form of ice VI referred to as ice XIX. With ice XV, this would be the second H-ordered form of ice VI, and a counter-example of the presently observed bijective relation between H-disordered and H-ordered phases. Ice XIX is located in the same P-T region as a previously reported variant of ice XV denoted beta-XV [Ref. 13]. Compared to this previous work in which ice samples were obtained at high pressures and then characterized on pressure-quenched samples, the present authors performed in situ studies. My general opinion is that although the experimental data appear solid and point to a different structure of ice than ice XV, the present work does not constitute a major progress from what was already known from the previous study cited above which would justify its publication in Nat. Comm. I also believe that before claiming a new phase of ice and assigning a Roman numeral to it, its structure should be firmly established, which could not be achieved in the present study. In addition, I find that the discussion of the dielectric data is poor compared to the rather exhaustive analysis of the neutron data. Detailed comments are given below:

- The quantitative analysis of dielectric data is poor: only the shift of the loss peak intensity with T is reported. What about the relaxation time and related activation energy which can be inferred from the peak frequency ? There is an apparent change of slope sign at 1.6 GPa (from negative to positive) in the loss peak intensity vs T evolution in the ice VI domain (Fig. 2b and Fig. S2), which is not commented at all. Dielectric data on non-doped H₂O is reported in SM but not mentioned in main

text and are barely commented, although clear differences with the data on doped ice are apparent. How do the authors explain these differences ?

- In figure S1a, the authors report the loss peak intensity from 124 K to 150K. However from Figure S1b the peak at 124 K is clearly not in the measured frequency window, how did the authors obtained its intensity ? It would also be useful to the readers if the authors gave the complete set of dielectric data they measured at each pressure in the SM.

- The authors define the disorder-order phase transition as the temperature at which the slope of the loss peak intensity changes. What is the physical basis for this definition ? Similarly, the author state that the transition is first order, however both dielectric signals and lattice parameters appear to change continuously over a temperature range of ~ 10 K, which at first-sight is not compatible with a first-order process. It is possible that the continuous character arises from the coexistence of the two phases and kinetics of the transition but this has to be discussed.

- Comparison with the data on beta-XV phase reported by Gasser et al [Ref. 13] is absent. How do the authors's dielectric data compare with those reported in this work? Do the authors think that their ice XIX is the same as Gasser et al's ice beta-XV, and if not why ? Did they try to recover samples of ice XIX at room pressure and characterize them as in Ref. 13 ?

- There are little discussion on kinetic effects in the experiments, which are known to be very important in the low T regime of the present experiments. In particular, the cooling rate in dielectric experiments is not specified. Did the authors perform measurements at different cooling rates ?

- The data is presented as isobaric scans. Can the authors explain how pressure was kept constant upon varying temperature ?

- The authors say that they did not consider the 5 lowest symmetry space group in their neutron powder refinement as "sufficient refinement agreements" were obtained for the other 18. They should specify what they mean by sufficient agreement, preferably in a quantitative manner.

- The authors say "centrosymmetry of hydrogen configurations is the most significance difference in hydrogen configuration between ice XIX and ice XV". But they found that P-4 is one of the most plausible space group of ice XIX, which to my understanding is a centrosymmetric group.

- As a minor comment, I find the title expression "new diversity form of ice polymorphism" rather odd and unclear. I had to read the abstract to understand its meaning. I suggest to remove it or find a better one.

Answer to reviewers' comments

#Reviewer 1

We appreciate Referee 1's helpful comments. Firstly, we will mention the modifications regarding the phase boundary between ice VI and its hydrogen-ordered phases, ice XV and XIX. This would be the main concern of Reviewer 1, as commented below:

1.1 In summary, on this point, although I don't claim my interpretation is more valid than the authors, I believe it illustrates a weakness in their volume argument and the location of the transition. In particular, the data should be presented with a serious analysis of systematic and random errors that could affect the transition temperatures extracted in this way. The proposal of a transition and, especially, the volume change have to be critically judged in light of this.

We reconsidered the definition of the phase transition temperatures, which was previously defined as the intersection of two straight lines fitted to the temperature dependence of dielectric loss peak intensity (DLPI) derived from ice VI and its hydrogen-ordered phases in the original submission. As indicated by Reviewer 1, the previous definition includes arbitrariness depending on how the two fitting regions are divided, we reanalyzed the temperature dependence of DLPI by the least arbitrariness, as shown from the following paragraph. Additionally, concerning the lower temperature region in which hydrogen ordering occurs, there is uncertainty for the previously employed linear approximation due to insufficient data points (this was also indicated by Reviewer 1). In the new definition, the DLPI of ice VI is assumed to be linearly dependent on temperature, and the phase transition temperature is redefined as the temperature at which the DLPI starts to deviate from the linearity. This deviation is caused by the hydrogen-ordering of ice VI as mentioned in our original manuscript. The following is the procedure for determining the deviation, taking the case of 1.9 GPa as an example. The raw DLPI data in the cooling run at 1.9 GPa are shown below (Figure 1-1; in this letter, we named figures by Figure "reviewer comment number-sequential number").

Figure 1-1| Temperature dependence of dielectric loss peak intensity (DLPI) of HCl-doped ice VI and its hydrogen-ordered phase (ice XIX) at 1.9 GPa upon cooling

1. First, it is obvious that the hydrogen ordering transitions occurred above 122 K in the cooling run. From this temperature, the DLPI data were obtained at several temperature points. In the case of the cooling run, {122, 124, 126, 128, 130, and 132} were selected. Hereafter, the number of selected temperature points is denoted by N .

2. Let us consider a temperature set selected in the same manner of step 1. We represent an element of the temperature set by T_i , where i takes from 1 to N and $T_i < T_{i+1}$. Linear fitting is conducted for the DLPI data, $\{d_i, \dots, d_{\max}\}$, obtained in the temperature range from T_i to the highest temperature of the measurement (T_{\max} ; T_{\max} was 150 K at 1.9 GPa). The fitted DLPI data are denoted by $\{\hat{d}_i, \dots, \hat{d}_{\max}\}$.

3. We calculate the residual sum of squares (RSS) between the observed $\{d_i, \dots, d_{\max}\}$ and the fitted $\{\hat{d}_i, \dots, \hat{d}_{\max}\}$, and then the RSS is normalized by the number of the DLPI data, $\{d_i, \dots, d_{\max}\}$. Hereafter, the normalized RSS is denoted by R_i . Let us consider that we decrease i from N (corresponding to a decrease of temperature T_i). When the hydrogen ordering happens, the normalized RSS should become large compared to that obtained above transition temperature due to the deviation from the linearity. This behavior can be shown in the following figure at around 126 K in the cooling run

(Figure 1-2).

Figure 1-2| Temperature dependence of normalized RSS in the cooling run at 1.9 GPa

4. Finally, to judge the phase transition temperature, we evaluate R_i/R_{i+1} (i takes from 1 to $N-1$). The figure below shows the temperature dependence of the ratio R_i/R_{i+1} obtained at various pressures (Figure 1-3). In this figure, the values of the ratio are nearly 1 in higher temperature region where ice VI is stable. This is because fitting residuals of R_i and R_{i+1} take similar values owing to good linearity between DLPI and measured temperature.

Figure 1-3 | Temperature dependence of the ratio R_i/R_{i+1} obtained in the pressure range 0.88–2.2 GPa upon cooling.

Based on the results, Figure 1-4 shows the phase diagram of ice VI and its hydrogen-ordered phases, determined using the ratio R_i/R_{i+1} of 1.5, 2, and 3 of as criteria values for the phase transition (Figure 1-4). The transition temperature is determined by $(T_i + T_{i+1})/2$, whose R_i/R_{i+1} is first above the criteria with decreasing i

of T_i from N . The displayed error bars show the temperature range from T_i to T_{i+1} . The three phase diagrams are only slightly different, and the main feature of the negative/positive dT/dP slope of ice VI and XV/XIX phase boundaries is common to all criteria. The relatively low transition temperature at 2.0 GPa might be caused by supercooling, which is a feature of first-order phase transition. In this study, the ratio, 2, was chosen as the criterion, because the criterion of 1.5 is occasionally too strict for the ratio obtained before the transition temperature; for example in the data of 1.6 GPa, the ratio is 1.42 and 1.37 at 122.5 and 125 K, respectively (Figure 1-3). If the criterion is too strict, the phase transition temperature would be overestimated. In addition, the ratio, 3.0, may underestimate the transition temperature considering such as the case of 1.7 and 2.0 GPa (see Figure 1-3). In the phase diagrams, the provisional phase boundary between ice XV and XIX is denoted between two pressures where the transition temperature increases (decreases) with increasing (decreasing) pressure. Also, transition temperatures of DCI-doped deuterated ice VI are shown. The D₂O samples show the compatible result with that obtained in H₂O ice, and also exhibit isotope effect in their phase transition temperatures, between which the difference is about 2 K. Such slight isotope effect has been reported in Ice I_h/XI and V/XIII phase transition (Koster et al., Phys. Rev. B **94**, 184306 (2016)), although we have no explanation for the isotope effect.

Figure 1-4| Phase diagrams of ice VI and its hydrogen-ordered phases obtained from the several criteria.

Based on the modification, we changed Figure 1 and Figure 2b in the main text, which show the modified phase diagram and temperature dependence of DLPI obtained in the pressure range 0.88-2.0 GPa with their transition temperatures. The modified figures are shown below. Note that in Figure 1, error bars are within the symbols showing phase transition temperatures. In Figure 2b, we previously showed the temperature dependence of DLPI obtained at 1.3 GPa, but the data are not shown in the modified figure due to the small number of data points compared to other pressures. The analysis of the phase transition temperature from ice VI to its hydrogen-ordered phases was added in Supplementary material.

Figure 1 | Representative experimental paths of dielectric and neutron diffraction experiments described in the phase diagram of ice obtained herein.

Figure 2 | Temperature dependence of dielectric properties of HCl-doped ice VI and its hydrogen-ordered phases.

Hereafter, based on the above discussion, we reply Referee 1's comments in order.

1.2 The first query I had relates to the colouring of the markers indicating the transition temperatures in Fig 1 and designating the phase: what determines which phase is which? Especially in the case of the 1.6 GPa point this appears to artificially lower the transition temperature. If this single point is questionable, then it seems to significantly reduce the evidence for a positive slope for the transition temperature in the proposed XIX-VI region.

In the modified determination of the transition temperatures, the coloring was changed between 1.5 and 1.6 GPa from which the transition temperature increases with increasing pressure. Additionally, in this revision, we have added a new figure to supplementary material regarding the temperature dependences of relaxation times following the advice from Reviewer 3. This figure would also resolve the concern that “*the 1.6 GPa point this appears to artificially lower the transition temperature*”. As known in other hydrogen ordering of ice, such as from ice Ih to XI, the relaxation time becomes longer accompanied with the phase transition due to the suppression of molecular reorientation. We observed this phenomenon in the case of ice VI (see the below Figure 1-5). In the figure, the phase transition from ice VI to ice XIX can be seen more obviously than our DLPI result. Also, the phase transition observed at 1.6 GPa occurs more firmly at the lowest temperature than at the higher pressures. It is noted that the relaxation time reflects the hydrogen ordering at a slightly lower temperature than DLPI. This could be related to the difference in the degree of domain growth of the ordered phase to change relaxation time and DLPI. On the other hand, it is difficult to determine the phase transition between ice VI and XV from the temperature dependence of relaxation time due to its slight change accompanied by the hydrogen ordering. This is the reason why we did not use the data of relaxation times for the phase-boundary analysis. We added the analysis of dielectric relaxation to Supplementary Material (content 1).

Figure 1-5 | Temperature dependence of relaxation time of HCl-doped ice VI (a) and DCI-doped deuterated ice VI (b). The red lines were fitted based on the Arrhenius equation and their activation energies are also shown.

1.3 If I were instead to consider fitting a straight line to the high temperature data, only using points in a region well away from the transition, then I could get quite a different behaviour. In the case of the 1.6 GPa data point, I might then put the transition point at the red arrow below and leading to a transition temperature that is more like the red-dashed line: This would lead to a quite different interpretation, where there is a phase boundary around 1.4 GPa (incidentally, coinciding closely with the apparent transition temperature reported by Glasser et al of 1.45 GPa), then a horizontal transition temperature (consistent with the authors observation of no volume change from diffraction) up to the last data point at 2.2 GPa at which, clearly, the transition temperature changes. However, could not the change of the 2.2 GPa data point be related to the approach to an ice VIII boundary?

Also, related to the following comment:

I would have liked to see the data at 2.2 GPa and it's not clear why this doesn't appear in either the main manuscript or the supplementary material. It would surely be important to show that the same peaks were observed as at 1.6 GPa?

As suggested by Referee 1, we added the neutron diffraction patterns obtained at 2.2 GPa in supplementary material as shown below (Figure 1-6). The diffraction patterns also show obvious change along with the hydrogen ordering, which occurs at a higher temperature compared to the diffraction pattern obtained at 1.6 GPa. Two important new peaks were observed at around 2.2 Å (indicated by blue ticks), which is the evidence that ice XIX has a distinct crystal structure. It should be mentioned that ice VIII (indicated by a black tick) coexists under the pressure. Since ice VIII already appears from 150 K and the existence of ice VIII would not affect the Gibbs's energy of ice VI and XIX (in other words, their transition temperature), it would be reasonable to consider that the higher transition temperature at 2.2 GPa is a consequence of the positive dT/dP slope.

Figure 1-6| Comparison of neutron diffraction patterns obtained at 1.6 and 2.2 GPa. Blue ticks indicate new peaks at 2.20 Å and 2.26 Å, which cannot be explained by the unit cell of ice XV. The peak indicated by the black tick is derived from ice VIII.

Very recently, Gasser et al. seem to have submitted a similar report to a Nature journal (published as a preprint in Research Square, DOI: 10.21203/rs.3.rs-86075/v1). In their report, they show powder neutron diffraction of deuterated samples decompressed to ambient pressure. Their diffraction pattern is well consistent with our obtained one. In their manuscript, the volume contraction is reported along with the phase transition from ice VI to ice XIX. Taking account of the Clausius–Clapeyron relationship, i.e. $dT/dP = \Delta V/\Delta S$, their result supports our analysis of phase boundary. Although the manuscript by Gasser et al. would be complementary with ours, it should be stressed

that our high-pressure study genuinely clarifies hitherto unknown part of the ice phase-diagram where the existence of two different structures by different hydrogen ordering has been first discovered.

1.4 I would also have been interested to see data at the same temperature, but lower pressure: the absence of these new peaks would strengthen the evidence for the location of the phase transition. This would be helpful especially as the DLPI data for the D₂O sample seems less clear than for H₂O.

Although we do not have the diffraction data obtained at the lower pressure where ice XV is stable as a hydrogen-ordered phase of ice VI, Komatsu et al. (Sci. Rep. **6**, 28920 (2016)) reported neutron diffraction pattern obtained in a similar temperature region at 0.4 GPa. From their results, we can confirm the absence of the new peaks appearing along with the hydrogen ordering from ice VI to XIX.

1.5 I noted that in the neutron data, the new peaks appear between 108 and 115 K, but this seems to be cooler than the transition temperatures given by the DLPI. I wondered if this was an isotope effect (the D₂O DLPI data seem less reliable than the H₂O so hard to say), or whether it might also be an effect of cooling rate?

The difference in sensitivity of the measurements for hydrogen-ordering would cause the indicated gap. To observe hydrogen-ordering from appearance of neutron diffraction peakss, larger domain growth may be needed, whereas lattice parameter or dielectric measurements are much more sensitive to the hydrogen-ordering. The phase boundary derived from the temperature dependence of lattice parameter (Fig. 3b in the main manuscript) shows consistent phase transition temperature at around 118 K with that obtained from the DLPI data (Figure 1-4), meanwhile significant intensity change cannot be observed in the diffraction patterns even at 115 K. Similar phenomena are observed in the case of ice VII-VIII phase transition (Komatsu et al., PNAS, **117**, 6356-6361 (2020)).

1.6 The authors conducted a quite thorough investigation of possible model fits to the neutron data and this is perhaps the most valuable work in the paper. I noticed that the consistent misfits of the peak width at $\sim 1.81 \text{ \AA}$ seem to be due to a large, Lorentzian-like broadening of the measured peak. Since such a peak-shape is often associated with particle-size effects, I wondered if this could be due to finite domain sizes of the ordered

phase. In this case, they would be quite directional as only this peak is affected. Maybe this is worth further comment.

We agree that the peak broadening was caused by particle-size effects (this point was mentioned in **Methods** of our original manuscript). Although the peak at ~ 1.81 Å seems to have a strong peak broadening, this degree of broadening can be also seen in other new peaks. The reason for the apparently significant broadening of the peak at ~ 1.81 Å would be just its strong intensity as compared to the other new peaks.

1.7 Lastly, on the refinements, there were several places where it was mentioned that the refinements were “conducted several times for each model” and the subsequent χ^2 values “averaged values over several refinement results”. This seemed an unusual process to me, as I would expect a given model to converge to the exact same structure when refined to the same data.

We changed the sentence “conducted several times for each model” to a more direct one shown in the following to avoid the doubt as indicated by Reviewer 1.

In **structure analysis of ice XIX** of **Methods** and also a part of the caption in Supplementary Fig. 6 (the following modification is about the caption of Supplementary Fig. 6 as a representative example):

Before) Structure refinements were conducted several times for each model to confirm their reproducibility; these results are plotted in this figure.

↓ (The underlined part was changed.)

After) In the first step of structure refinements, the site occupancies of hydrogen atoms were fitted one by one, and subsequently fitted together as variables. Since the first step has arbitrariness in its fitting order (e.g. $\alpha \rightarrow \beta \rightarrow \dots$ and $\beta \rightarrow \alpha \rightarrow \dots$), we conducted structure refinements in several ways for each model by changing the fitting order cyclically, such as $\alpha \rightarrow \beta \rightarrow \dots$ and $\beta \rightarrow \gamma \rightarrow \dots$. However, the fitting results are almost independent of the order.

1.8 It would have been helpful in the review to have had a cif file and a secondary check on the model would have been whether sensible O-D bondlengths were obtained (coordinates are given, but I'm afraid I didn't have time to enter all of these manually). Also, were these freely refined or were constraints used?

We added tables listing the O-D bond lengths in Supplementary Material as Supplementary Table 2 and 4. Our structure refinement employed no constraint for atomic positions. Cif files of the structure models, $P\bar{4}$ and $Pcc2$, will be deposited at the Cambridge Crystallographic Data Centre (CCDC) if our manuscript is published.

1.9 line 90: “the phase boundary between ice XV and XIX should have a slope rather than lie horizontally as suggested previously”. I found this confusing as the phase diagram the authors use has pressure on the x-axis and temperature on the y-axis. In this case, the transition between XV and XIX should be approximately vertical not horizontal?

We appreciate this indication and changed the sentence as shown below.

Before) In this context, the phase boundary between ice XV and ice XIX should have a slope rather than lie horizontally as suggested previously, because ice XV has a larger volume than ice XIX (the supposed phase boundary in Fig. 1 is shown vertically to emphasise this point).

↓ (The underlined part was changed.)

After) In this context, the phase boundary between ice XV and XIX would be close to a vertical unlike horizontal suggested in the previous study if their entropy difference is enough small, because ice XV has a larger volume than ice XIX (the supposed phase boundary in Fig. 1 is shown vertically to emphasise this point).

1.10 line 156 and 162 attempt to reconcile DSC measurements on recovered samples, interpreted in terms of a “deep-glassy state of ice VI” with their observation of a new, ordered, crystalline phase. The authors seem to suggest that their ordered phase may disorder upon pressure lowering. This, to me, seemed a little counter intuitive: even if hydrogen mobility increases as pressure reduces, where would the additional energy come from to disorder the dipoles, which are already in an lower energy ordered state?

Thermal fluctuation might suppress the long-range order of ice XIX, but in the experimental time scale, the fluctuation is not enough to cause a phase transition from ice XIX to more stable hydrogen-ordered phase of ice VI under ambient pressure. In this context, the suppression would cause an amorphous-like structure rather than a disordered state. This amorphization is well known in decompressed samples to ambient

pressure as a phase transition from a high-pressure phase to an amorphous phase.

1.11 The authors should provide some further justification for their explanation. line 183 mentions “large water molecules” what does this refer to? Aren't all water molecules the same size?

Thank you so much for the suggestion; “large water molecules” was corrected to “a large number of water molecules”.

1.12 Supplementary line 99: From the description of the pressure measurement, via ruby fluorescence, it seemed quite plausible to me that there would be a systematic deviation between ruby pressure and sample pressure, due to pressure gradients and that the ruby lies at one end of the sample. Given that the uncertainty in transition pressures of a few kbars would have a significant effect on the drawing of phase boundaries, did the authors attempt to quantify this and (if necessary) apply a correction?

We deem that the pressure correction is not necessary. Although we have not confirmed pressure gradient in our developed cell, piston-cylinder apparatus would only have enough small pressure gradient compared to its general achievable pressure, about 2 GPa, unless samples are compressed/decompressed at low temperature, such as 77 K. It is noted that in our dielectric measurements every compression was conducted at room temperature. In addition, even if there is a pressure gradient, the ruby is always set at the same position, as shown in the cell assembly (Supplementary Figure 10). The pressure discrepancy between sample and ruby, if any, becomes only offset of our pressure estimation. This would not affect our main claims that ice VI has two types of hydrogen ordering depending on pressure and the hydrogen-ordered phases show opposite volume change each other in the phase transition from ice VI.

#Reviewer 2

We appreciate Reviewer 2's positive response, and completely agree his/her indication for our structure refinements of the new hydrogen-ordered ice XIX. As indicated by Reviewer 2, it would be difficult to determine the crystal structure of the new phase from powder neutron diffraction, and we would be happier if the crystal structure is strictly solved using single crystal samples overcoming its experimental difficulties. Also, we hope that the new hydrogen-ordering phenomena and our new technical device development inspires many studies in the wide-ranging fields e.g., geosciences, physics, physical chemistry, crystallography, and so on.

#Reviewer 3

3.1 My general opinion is that although the experimental data appear solid and point to a different structure of ice than ice XV, the present work does not constitute a major progress from what was already known from the previous study cited above which would justify its publication in Nat. Comm.

For the first time, we have established the presence and nature of a new hydrogen-ordered ice which shows a different hydrogen-ordering as compared to the already known hydrogen-ordered phase (ice XV). Although the existence of the new hydrogen-ordered phase is supposed by Gasser et al. as β -XV, no direct evidence has been obtained thus far. Here we show the first, unambiguous, structural evidence for the new phase (ice XIX) by in-situ neutron diffraction under high-pressure. In addition, our high-pressure dielectric measurement clarifies that the hitherto unknown phase diagram of ice VI and its hydrogen-ordered phases which should be essential information for our deep understanding of the intriguing hydrogen-ordering of ice VI. The new phase diagram of ice will directly inspire many studies in the wide-ranging fields, e.g., geosciences, physics, and physical chemistry. It is also stressed the point that our newly developed high-pressure cell for the dielectric measurements would be a powerful tool for further investigation of the various hydrogen-ordering of ice in detail as with this study.

3.2 In addition, I find that the discussion of the dielectric data is poor compared to the rather exhaustive analysis of the neutron data. Detailed comments are given below:

We added a more detailed analysis for dielectric measurements as shown below.

3.3 The quantitative analysis of dielectric data is poor: only the shift of the loss peak intensity with T is reported. What about the relaxation time and related activation energy which can be inferred from the peak frequency ?

We added the analysis of dielectric relaxation in Supplementary Material (content 1) which is shown in the below Figure 3-1 (we here named figures by Figure “reviewer comment number-sequential number”). As known in other hydrogen ordering of ice, the relaxation time becomes longer when the phase transition occurs. This is due to the suppression of molecular reorientation. We observed this behavior in the phase

transition between ice VI and XIX (Figure 3-1). Activation energy of HCl-doped ice VI and DCI-doped deuterated ice VI is about 0.2 eV, which is a consistent value with that of other HCl/DCI-doped and also KOH/KOD-doped disordered ice, such as ice V and I_h (Koster et al., Phys. Rev. B **94**, 184306 (2016); Kawada, J. Phys. Soc. Jpn. **58**, 295-300 (1989)).

Figure 3-1 | Temperature dependence of relaxation time of HCl-doped ice VI (a) and DCI-doped deuterated ice VI (b). The red lines were fitted based on the Arrhenius equation and their activation energies are also shown.

On the other hand, the phase transition between ice VI/XV and DCI-doped ice VI/XIX show a relatively small change in their relaxation times. The previous dielectric study on ice V and I_h reported a similar isotope effect in their hydrogen (heavy hydrogen) ordering. Although the reason for this difference is not yet clear, Kawada (1989) indicated that the degree of hydrogen/heavy hydrogen ordering would be related to the difference. As with the isotope effect, the difference in the degree of hydrogen-ordering might also be the reason for the relatively small change in the relaxation time of the phase transition from ice VI to ice XV compared to that between ice VI and XIX. Figure 3-2 shows the temperature dependence of dielectric loss peak intensity obtained in cooling and heating runs at 0.88 and 2.2 GPa, where ice XV and XIX are stable, respectively. It can be seen that the hydrogen ordering of ice XV happens in a wider

temperature region (~ 20 K) compared to that of ice XIX (~ 5 K). These results are consistent with the supposed difference in the degree of hydrogen-ordering between ice XV and XIX. The difference would be due in part to the height of the activation barrier of hydrogen ordering and energy difference between ice VI and the hydrogen-ordered phases. However, we need further investigation for such discussion.

Figure 3-2 | Temperature dependence of dielectric loss peak intensity obtained in cooling and heating runs at 0.88 and 2.2 GPa, where ice XV and XIX is stable, respectively. At 2.2 GPa, data measured below 124 K (upon cooling) and 130 K (upon heating) are not shown here, because the dielectric response of ice XIX almost disappeared in the temperature region.

3.4 There is an apparent change of slope sign at 1.6 GPa (from negative to positive) in the loss peak intensity vs T evolution in the ice VI domain (Fig. 2b and Fig. S2), which is not commented at all.

As shown in the left side of Figure 3-3, the temperature shift of dielectric loss peak was obtained in two different high-pressure runs conducted at 1.9 GPa, where their samples were changed. These data show no reproducibility of the indicated tendency, such that

we did not mention anything about that. It is stressed the point that the different samples of ice VI show the consistent temperature dependence of relaxation times and phase transition temperature to ice XIX each other (see the right side of the figure).

Figure 3-3 Comparison of a temperature shift of dielectric loss peak of ice VI (left side) and temperature dependence of relaxation time of ice VI and XIX (right side) obtained in two different high-pressure runs conducted at 1.9 GPa, where their samples were changed.

3.5 Dielectric data on non-doped H₂O is reported in SM but not mentioned in main text and are barely commented, although clear differences with the data on doped ice are apparent. How do the authors explain these differences ?

It is reasonable that pure ice and HCl-doped ice exhibit different dielectric responses, because the chemical dopant locally breaks the ice rules by which the molecular reorientation can be activated. This activation causes clear differences in terms of the dielectric responsibility of ice. Quantitatively, pure ice VI has higher activation energy for the molecular reorientation, ~0.5 eV (measured at 1.1 GPa in Johari et al., 61, 4292-4300 (1974)), compared to that of the HCl-doped ice VI, ~0.2 eV (Figure 3-1). We added this explanation in Supplementary Material.

3.6 In figure S1a, the authors report the loss peak intensity from 124 K to 150K.

However from Figure S1b the peak at 124 K is clearly not in the measured frequency window, how did the authors obtained its intensity ? It would also be useful to the readers if the authors gave the complete set of dielectric data they measured at each pressure in the SM.

The raw data and fitted curves of the dielectric constant and loss are shown below in Figure 3-4, in which loss tangent is also displayed for reference. Although, as indicated by Reviewer 3, the dielectric loss peak is unclear in the temperature, we conducted its model fitting by also referring other dielectric properties, dielectric constant and loss tangent. We added the measured data of dielectric properties of ice VI and its hydrogen-ordered phases.

Figure 3-4| Raw data and fitted curves of the dielectric properties of ice XIX, dielectric constant, dielectric loss, and loss tangent, obtained at 124 K and 1.9 GPa.

3.7 The authors define the disorder-order phase transition as the temperature at which the slope of the loss peak intensity changes. What is the physical basis for this definition ?

Also, related to the next comment:

Similarly, the author state that the transition is first order, however both dielectric signals and lattice parameters appear to change continuously over a temperature range of ~10 K, which at first-sight is not compatible with a first-order process. It is possible that the continuous character arises from the coexistence of the two phases and kinetics of the transition but this has to be discussed.

First, the temperature hysteresis along with phase transition between ice VI and XIX (Supplementary Figure 1) is the evidence that the phase transition is first order. The indicated (apparent) continuous change is caused by the coexistence of the disorder/order phases. It is expected that first-order phase transition shows a sudden change in physical properties of samples; for example, dielectric properties and lattice parameters are expected to be changed in the case of hydrogen ordering of ice. In the dielectric measurements, we use dielectric loss to analyze the phase transition because the peak is easy to be traced to its temperature change. Although dielectric constant is generally used in disorder/order phase transition of dielectrics, the intensity of both physical quantities, dielectric constant and loss, is mainly changed by static dielectric constant ϵ_0 based on the Debye dispersion model. Therefore, the dielectric loss peak intensity is also appropriate for the evaluation of the phase transition.

3.8 Comparison with the data on beta-XV phase reported by Gasser et al [Ref. 13] is absent. How do the authors's dielectric data compare with those reported in this work? Do the authors think that their ice XIX is the same as Gasser et al's ice beta-XV, and if not why? Did they try to recover samples of ice XIX at room pressure and characterize them as in Ref. 13?

Gasser et al. showed dielectric loss data of the decompressed sample from 1.8 GPa to ambient pressure at 77 K (Gasser et al., Chem. Sci. **9**, 4224-4234 (2018)). We compared our dielectric loss data obtained at 1.9 GPa with the data reported by Gasser et al. (2018). There is an apparent difference between them (see Figure 3-5) in the temperature region below about 108 and 124 K, where the samples are in the hydrogen-ordered state, respectively. The difference might reflect the revival of molecular reorientation, which should be immobilized upon hydrogen ordering, in the decompressed sample. Taking account of the “deep-glassy state” suggested by Rosu-Finsen et al. (Chem. Sci. **10**, 515-523 (2019)), we consider that it is non-trivial that ice XIX and the decompressed sample are the same ones, although we do not have neutron diffraction data of such decompressed samples to confirm that.

Figure 3-5 | Comparison of the temperature dependence of dielectric loss data between the previous study (Gasser et al. *Chem. Sci.* **9**, 4224–4234 (2018)) and this study.

3.9 There are little discussion on kinetic effects in the experiments, which are known to be very important in the low T regime of the present experiments. In particular, the cooling rate in dielectric experiments is not specified. Did the authors perform measurements at different cooling rates ?

We agree that the systematic study for the kinetics effects on the hydrogen ordering of ice VI is important, but in this study, we did not conduct such experiments because our main purpose is to obtain direct evidence for the existence of the second hydrogen-ordered phase of ice VI. All dielectric measurements in this study were conducted using the same cooling rate 2 K/h.

3.10 The data is presented as isobaric scans. Can the authors explain how pressure was kept constant upon varying temperature?

We did not keep the pressure constant. The figures below plot sample pressures determined in dielectric and neutron diffraction measurements at 1.9 and 1.6 GPa,

respectively. The sample pressure slightly changed with decreasing (increasing) temperature.

Figure 3-6 | Representative temperature-dependence of sample pressure in dielectric (a) and neutron diffraction (b) measurements at 1.9 and 1.6 GPa, respectively.

We mentioned the sample-pressure change in **dielectric measurements** in **Methods** as following:

It is noted that sample pressure is slightly changed by decreasing (increasing) temperature at most about 0.1 GPa in the measured temperature region. The shown pressure of the dielectric data corresponds to that measured at around phase-transition temperature. In the neutron diffraction measurement, the sample pressure also changed about 0.1 GPa in the measured temperature range, and the shown pressure was determined in the same manner as the dielectric measurement.

3.11 The authors say that they did not consider the 5 lowest symmetry space group in their neutron powder refinement as "sufficient refinement agreements" were obtained for the other 18. They should specify what they mean by sufficient agreement, preferably in a quantitative manner.

We agree that the sentence "sufficient refinement agreements" is not an appropriate representation and modified it as shown below.

Before) We conducted Rietveld analyses using structural models with 18 space groups of the remaining candidates, except for the lower-symmetry space groups: Pc , $P2_1$, $P2$,

$P\bar{1}$ and $P1$ —this cut-off is based on indices of the subgroups of $P4_2/nmc$ (see details in Supplementary information). Notably, we do not rule out the possibility that the actual crystal structure of ice XIX having one of these space groups, although **sufficient refinement agreements** were obtained for the 18 candidates from our neutron diffraction data.

After) We conducted Rietveld analyses using structural models with 18 space groups of the remaining candidates, except for Pc , $P2_1$, $P2$, $P\bar{1}$ and $P1$, which are the lower-symmetry space groups of $P\bar{4}$, $Pca2_1$, $Pcc2$, $P2_1/a$ and $P2_1/c$ selected as the best candidates for ice XIX based on their fitting χ^2 values. Since the best candidates show close χ^2 (between 5.3 and 6.0 as shown in Supplementary Fig. 8) and they can explain all observed new peaks of ice XIX (see Supplementary Fig. 9), the lower-symmetry space groups were not considered here. Notably, we do not rule out the possibility that the actual crystal structure of ice XIX having one of the lower-symmetry space groups.

Along with this modification, we added a new figure in Supplementary Material (Supplementary Figure 9) which shows finally fitted lines for the experimentally obtained neutron diffraction patterns of ice XIX using the $P\bar{4}$ and $Pcc2$ structure model.

Supplementary Figure 9 | Neutron diffraction patterns collected at 1.6 GPa and 80 K (black dots) and finally fitted lines (coloured by red) using the most plausible structure models for ice XIX, $P\bar{4}$ (a) and $Pcc2$ (b). The black ticks represent all the peak positions expected from the unit cells of ice XIX. The blue lines show residuals between the observed and simulated diffraction patterns.

3.12 The authors say "centrosymmetry of hydrogen configurations is the most significance difference in hydrogen configuration between ice XIX and ice XV". But they found that $P\bar{4}$ is one of the most plausible space group of ice XIX, which to my understanding is a centrosymmetric group.

Although $P\bar{4}$ does not have polar direction (Reviewer 3 perhaps imply this point), no centrosymmetry exists in the space group.

3.13 As a minor comment, I find the title expression "new diversity form of ice polymorphism" rather odd and unclear. I had to read the abstract to understand its meaning. I suggest to remove it or find a better one.

We agree to this comment and changed the title to “Ice XIX: Discovery of second hydrogen ordered phase of ice VI”.

REVIEWERS' COMMENTS

Reviewer #1 (Remarks to the Author):

The authors have clearly put a lot of work into addressing my comments. I thank them for this and am happy that the current form of the manuscript allows me to recommend its publication.

Reviewer #2 (Remarks to the Author):

Little mistake in the references: reference [6] must be "Kuhs, W.F., ..." rather than "W.F., K., ..." -- probably due to a mistake in the reference database, mixing up first name(s) and family name.

In the supplementary material, the black tick in Figure 1-6 is barely distinguishable from the two blue ones. Needs to be, e.g., red to green to be easily distinguishable.

Otherwise, I can not complain about the corrections the authors suggest, having been far less demanding as compared to reviewers 1 and 3. Nevertheless, considering previous work, notably by Gasser et al., I'd rather replace the exciting "discovery" in the title by "confirmation", "... Confirmation of a second hydrogen ordered phase..."

At the first submission of this manuscript, the authors could not have been aware of a very recent paper: Tobias Gasser, Alexander Thoeny, Andrew Fortes & Loerting, T. Ice XIX: The second hydrogen-ordered polymorph related to ice VI. Research Square, doi:10.21203/rs.3.rs86075/v1 (2020). It appeared after the first submission but before this revision (logically...). It had been submitted on the preprint server 7th of October, whereas this manuscript had been submitted for Nat. Comm. Taking into account this recent work would have an impact on virtually everything in the current manuscript and would delay its publication due to the necessary profound revision substantially. I therefore wonder whether it is worthwhile to cite the preprint (which is quite complementary to the manuscript, as reporting on an ex situ structure study, as compared to an in situ study here, but coming to a converging structural conclusion, with space groups P-4 and Pcc2 being the most likely candidates).

Reviewer #3 (Remarks to the Author):

This revision does not change my general opinion expressed in my first report: the experimental data appear solid and point to a different structure of ice than ice XV. However, a definite structural model of this new form could not be disclosed. I acknowledge that this is not an easy task and that it may take some time before conclusive data are obtained. I thus leave it to the Editor to decide whether the incremental knowledge that this paper brings is worth publication in Nat. Comm. As the authors, I became aware of a competing paper on the same subject and apparently submitted to the same journal. My recommendation would be to make a common decision regarding the publication of the two papers since the experimental data are similar and the conclusions are identical.

Some additional comments about the authors' response and revised manuscript are:

- As noted by other referees, I find it far-stretched to call ice XIX an "ordered" phase of ice since in the two finally selected models, several deuterium sites have occupancy factors of 0.5, which clearly implies hydrogen disorder. A better term would thus be "partially ordered". Similar conclusions have been drawn for ice XV (Komatsu et al, Sc. Rep. 2016).

- I appreciate that the authors included the data on the dielectric relaxation time. The increase in relaxation time at the disorder-order transition appear clear for data above 1.6 GPa and much less so for the data below, as noted by the authors themselves. The authors' arguments in an attempt to explain this are not fully clear to me (some rephrasing is needed), but from what I understand they relate this to a lesser degree of ordering in ice XV than in ice XIX. This adds to the comment above that these phases cannot be called "ordered" phases. I also note that the relaxation data below 1.6 GPa appear less consistent with each other in the "ordered" phase. Could this be due to experimental artefacts such as pressure variation or to noisier data at low pressure?

- I noticed that a few O-D distances in the refined models are too short by up to 0.19 Ang than the refined value of 0.968(7) Ang value in ice VIII at 2.4 GPa (Kuhs et al, JCP 1984), which is unlikely. Did the authors try to constrain the O-D distance to this value and how much did this worsen the fit?

Answer to reviewers' comments

#Reviewer 1

1.1 The authors have clearly put a lot of work into addressing my comments. I thank them for this and am happy that the current form of the manuscript allows me to recommend its publication.

We appreciate again your indications and advice. Your suggestions greatly helped us revise our manuscript to make it more convincing.

#Reviewer 2

2.1 Little mistake in the references: reference [6] must be "Kuhs, W.F., ..." rather than "W.F., K., ..." -- probably due to a mistake in the reference database, mixing up first name(s) and family name.

We appreciate this indication and modified the reference as below:

5. Kuhs, W. F., Finney, J. L., Vettier, C. & Bliss, D. V. Structure and hydrogen ordering in ices VI , VII , and VIII by neutron powder diffraction. *J. Chem. Phys.* **81**, 3612 (1984).

2.2 In the supplementary material, the black tick in Figure 1-6 is barely distinguishable from the two blue ones. Needs to be, e.g., red to green to be easily distinguishable.

We changed the colors using red and blue as below:

Supplementary Figure 5| Comparison of neutron diffraction patterns obtained at 1.6 and 2.2 GPa. Blue ticks indicate new peaks at around 2.2 Å, which do not appear from the unit cell of ice XV. The peak indicated by the red tick is derived from ice VIII.

2.3 Nevertheless, considering previous work, notably by Gasser et al., I'd rather replace the exciting "discovery" in the title by "confirmation", "... Confirmation of a second hydrogen ordered phase..."

We agree to this comment and changed the title to “Experimental evidence for the existence of a second partially-ordered phase of ice VI”.

2.4 At the first submission of this manuscript, the authors could not have been aware of a very recent paper: Tobias Gasser, Alexander Thoeny, Andrew Fortes & Loerting, T. Ice XIX: The second hydrogen-ordered polymorph related to ice VI. Research Square, doi:10.21203/rs.3.rs86075/v1 (2020). It appeared after the first submission but before this revision (logically...). It had been submitted on the preprint server 7th of October, whereas this manuscript had been submitted for Nat. Comm. Taking into account this recent work would have an impact on virtually everything in the current manuscript and would delay its publication due to the necessary profound revision substantially. I therefore wonder whether it is worthwhile to cite the preprint (which is quite complementary to the manuscript, as reporting on an ex situ structure study, as compared to an in situ study here, but coming to a converging structural conclusion, with space groups $P\bar{4}$ and $Pcc2$ being the most likely candidates).

Since the editor informed us that “*We are planning to publish the two works on the same day.*”, we cited the other complementary paper to support this study (especially for phase diagram and structural analysis of ice XIX) as follows:

“It is noted that Gasser et al. also came to the identical conclusion from their neutron diffraction measurements using decompressed samples at about the same time as we did. They referred our structural analysis and reported that the best fitting was obtained by the $P\bar{4}$ and $Pcc2$ structural models, the same as our results²⁰. They reported volume contraction of the hydrogen ordering from ice VI to XIX at ambient pressure. This result supports the phase diagram of ice VI and its hydrogen ordered phases clarified in this study.”

#Reviewer 3

3.1 As noted by other referees, I find it far-stretched to call ice XIX an "ordered" phase of ice since in the two finally selected models, several deuterium sites have occupancy factors of 0.5, which clearly implies hydrogen disorder. A better term would thus be "partially ordered". Similar conclusions have been drawn for ice XV (Komatsu et al, Sc. Rep. 2016).

We agree to the indication from Reviewer 3 regarding the partially ordered state of ice XIX. We clarified this fact in our title, abstract and also main text by adding the following words and sentences:

Title) Experimental evidence for the existence of a second partially-ordered phase of ice VI

Abstract) Here we report a high-pressure phase, ice XIX, which is a second hydrogen-partially-ordered phase of ice VI.

Main text) In addition, the $P\bar{4}$ or $Pcc2$ structural models include deuterium (hydrogen) atoms whose site occupancy is 50 %; in other words, the models indicates that ice XIX is partially ordered state as with the $Pmnm$ structure model suggested for ice XV by Komatsu et al.

3.2 I appreciate that the authors included the data on the dielectric relaxation time. The increase in relaxation time at the disorder-order transition appear clear for data above 1.6 GPa and much less so for the data below, as noted by the authors themselves. The authors' arguments in an attempt to explain this are not fully clear to me (some rephrasing is needed), but from what I understand they relate this to a lesser degree of ordering in ice XV than in ice XIX. This adds to the comment above that these phases cannot be called "ordered" phases. I also note that the relaxation data below 1.6 GPa appear less consistent with each other in the "ordered" phase. Could this be due to experimental artefacts such as pressure variation or to noisier data at low pressure?

As you mentioned, we deem that ice XV has less degree of hydrogen-ordering than ice XIX. If, as indicated in Komatsu et al, *Sci. Rep.* 2016, ice XV has the $Pmnm$ structure model, both of the phases, ice XV and XIX, should be called partially hydrogen-ordered phases. In addition, due to the less degree of hydrogen order of ice XV, its domain

growth might be more easily affected by external factors (such as thermal fluctuation, the way of introduction of dopant, and so on.). This could cause the indicated less consistent relaxation data of ice XV.

3.3 I noticed that a few O-D distances in the refined models are too short by up to 0.19 Ang than the refined value of 0.968(7) Ang value in ice VIII at 2.4 GPa (Kuks et al, JCP 1984), which is unlikely. Did the authors try to constrain the O-D distance to this value and how much did this worsen the fit?

We did not use any constraint for the O-D distance previously. In the revised manuscript, O-D bond lengths were constrained to be 0.95, which is an averaged value of the O-D bond length of the ice VI structural model, based on the neutron diffraction pattern obtained at 1.6 GPa and 80 K. Even under this constraint, $P\bar{4}$ and $Pcc2$ are the most plausible structure models. We confirmed that their O-D bond lengths are in the region of 0.93-0.96 Å.